# Treatment and Outcomes of Radiation-Induced Soft Tissue Sarcomas of the Extremities and Trunk—A Systematic Review of the Literature

**DOI:** 10.3390/cancers15235584

**Published:** 2023-11-25

**Authors:** Maria L. Inchaustegui, Kelly Kon-Liao, Kim Ruiz-Arellanos, George Aquilino E. Silva, Marcos R. Gonzalez, Juan Pretell-Mazzini

**Affiliations:** 1Facultad de Medicina, Universidad Peruana Cayetano Heredia, Lima 15102, Peru; maria.inchaustegui@upch.pe (M.L.I.); kelly.kon@upch.pe (K.K.-L.); kim.ruiz.a@upch.pe (K.R.-A.); 2Biology Department, Boston University, Boston, MA 02215, USA; georgeaquilinoesilva@gmail.com; 3Division of Orthopaedic Oncology, Department of Orthopaedic Surgery, Massachusetts General Hospital, Harvard Medical School, Boston, MA 02115, USA; mgonzalez52@mgh.harvard.edu; 4Division of Orthopedic Oncology, Miami Cancer Institute, Baptist Health System South Florida, Plantation, FL 33324, USA

**Keywords:** radiation-associated sarcoma, soft tissue sarcoma, surgery, survival

## Abstract

**Simple Summary:**

Radiation-induced soft tissue sarcomas (RISs) are rare cancers with a dire prognosis caused by past radiation therapy. Due to their low recurrence, they are poorly understood. The aim of this systematic review was to analyze how RIS is treated, and the outcomes that patients face. After reviewing 21 studies with 1371 RIS patients, it was found that surgery was the most common treatment, with chemotherapy and radiotherapy used less frequently. The most common histological type was undifferentiated pleomorphic sarcoma (42.2%). Patients with RIS had a 5-year survival rate of 45% and high rates of local recurrence (39%) and cancer spreading (27%). These findings shed light on the challenges of managing RIS and may guide future research to improve treatment outcomes for these patients.

**Abstract:**

Introduction: Radiation-induced soft tissue sarcomas (RISs) are rare secondary malignancies with a dire prognosis. The literature on the management of these tumors remains scarce due to their low incidence. Our systematic review sought to assess the treatment alternatives and outcomes of patients with RIS. Methods: A systematic review was conducted following the PRISMA guidelines. Our study was registered in PROSPERO (ID: CRD42023438415). Quality assessment was performed using the STROBE checklist. Weighted means for both continuous and categorical values were calculated. Results: Twenty-one studies comprising 1371 patients with RIS were included. The mean latency period from radiation to RIS diagnosis was 14 years, and the mean radiation dose delivered to the primary malignancy was 29.2 Gy. The most common histological type was undifferentiated pleomorphic sarcoma (42.2%), and 64% of all tumors were high-grade. The trunk was the most common location (59%), followed by extremities (21%) and pelvis (11%). Surgery was performed in 68% of patients and, among those with an appendicular tumor, the majority (74%) underwent limb-salvage surgery. Negative margins were attained in 58% of patients. Chemotherapy and radiotherapy were administered in 29% and 15% of patients, respectively. The mean 5-year overall survival was 45%, and the local recurrence and metastasis rates were 39% and 27%, respectively. Conclusions: In our study, the most common treatment was surgical resection, with RT and chemotherapy being administered in less than one third of patients. Patients with RIS exhibited poor oncologic outcomes. Future studies should compare RIS with de novo STS while controlling for confounders.

## 1. Introduction

Radiation-induced soft tissue sarcomas (RISs) are a devastating late complication of radiation therapy (RT). These neoplasms, which can originate in the bone or soft tissue, were first defined by Cahan et al. in 1948 [1]. In their study, they established the following diagnosis criteria: (1) history of radiation therapy; (2) occurrence of the sarcoma within a previously irradiated field; (3) asymptomatic latency period of several years; and (4) histologic confirmation of sarcomatous nature of the post-irradiation lesion [1]. With a 15-year cumulative incidence of 3.2 per 1000 people that have received RT, RISs are a rare group of neoplasms [2]. Nonetheless, incidence of RIS is increasing due to the longer survival of oncologic patients and more extensive use of RT within this population [3].

RISs typically present as large, high-grade tumors, and exhibit poorer oncologic outcomes compared to de novo soft tissue sarcomas (STSs) [4,5,6,7,8]. The management of RIS is similar to that of de novo STS and consists of surgical resection, with or without neo-adjuvant chemotherapy (QT). The role of RT in RIS, however, remains unclear due to potential side effects of reirradiation [9]. Due to the condition’s low incidence, comprehensive studies on the optimal treatment of RIS are scarce. Notably, there have been no large prospective studies or randomized controlled trials on the topic and the largest study currently available includes only 510 patients with RIS [10]. Moreover, the majority of the RIS literature primarily focuses on tumors arising in the chest wall after the irradiation of breast carcinoma [2,5,7,11,12]. To our knowledge, there have been no systematic reviews assessing the oncologic outcomes in this population.

In our systematic review, we sought to answer the following questions: (1) what are the demographic and clinical characteristics of patients with RIS? (2) What are the most common treatment strategies of patients with RIS? (3) What are the oncologic outcomes of patients with RIS?

## 2. Materials and Methods

### 2.1. Search Strategy

A comprehensive search of the PubMed and Embase libraries was conducted on 10 September 2023. The following terms and Boolean operators were used: (“radiation-induced” OR “radiation-associated” OR “post-irradiation” OR “postirradiation” OR “post-radiation” OR “postradiation”) AND sarcoma AND (bone OR soft tissue). In addition, we reviewed all included studies to identify references that may have been missed in our initial search. Our systematic review was registered in PROSPERO (ID: CRD42023438415).

### 2.2. Eligibility Criteria

To be included in our study, articles had to (1) report oncologic outcomes of RIS located in the extremities, trunk or pelvis, and (2) include at least ten patients diagnosed with an RIS in the aforementioned locations. Non-peer-reviewed publications and studies not written in English, Spanish, German, Italian or Portuguese were excluded.

### 2.3. Selection, Data Collection and Extraction 

A separate search query for each of the two databases used (PubMed and Embase) was performed. The results were then uploaded into Covidence^TM^ (Veritas Health Information) and duplicates were removed. Three reviewers (M.L.I., K.K.L. and G.A.S.) independently screened studies for eligibility. In case of disagreement, the senior author (J.P.M.) was consulted, and the final decision was reached by consensus.

Data from the included manuscripts were extracted into a pre-assembled spreadsheet. The following variables were retrieved: first author, year of publication, study design, country and institution where the study was performed, age and sex of patients, latency time and radiation dose (Gy), primary tumor radiated, follow-up, tumor location, tumor grade and size, histology, treatment characteristics and oncologic outcomes. Treatment variables retrieved included the percentage of patients receiving surgical management for RIS and the negative margin rate. For RIS located in the extremities, the type of surgery was classified as a limb-salvage surgery or amputation. RT and QT treatment strategy variables included both neoadjuvant and adjuvant therapies, given that most studies did not specify the distinction. 

For continuous variables like age and follow-up time, we collected mean or median values based on the reported metric used by the author. We evaluated three oncologic outcomes: five-year overall survival, local recurrence rate and metastasis rate. Five-year overall survival was determined based on the Kaplan–Meier estimates reported by each study. We defined local recurrence as the presence of tumoral tissue in the vicinity of the former RIS tumor bed or at the end of the amputation stump. Local recurrence and metastasis rates were calculated as the proportion of patients who experienced the event of interest at their last follow-up. We chose these outcomes because they were the most frequently reported in the included studies.

### 2.4. Study Selection and Characteristics

This systematic review followed the PRISMA guidelines. The search resulted in 1342 titles in PubMed and 525 titles in EMBASE, and the resulting datasets were exported to Covidence™ (Veritas Health Information). After duplicate removal, three independent reviewers (M.L.I., K.K.L. and G.A.S.) screened all titles and 1620 studies were excluded (Figure 1). The full manuscripts of 153 studies were then reviewed and a total of 22 articles met our inclusion criteria and progressed to quality assessment. 

### 2.5. Quality Assessment 

We evaluated study quality using the Strengthening the Reporting of Observational Studies in Epidemiology (STROBE) checklist. We employed 10 out of the 22 checklist items, consistent with prior orthopedic literature [13,14,15,16]. Each item received a score from 0 to 2, where well-described items earned 2 points, partly described items received 1 point and poorly described items obtained 0 points. Studies with a cumulative score of ≥15 points were considered for inclusion. We excluded 1 study due to a low score [17] and 21 studies were finally included in our systematic review (Figure 1). 

### 2.6. Caracteristics of Included Studies

A total of 1371 patients with radiation-induced sarcomas from 21 studies were included in our systematic review [3,4,5,6,7,8,9,10,11,12,18,19,20,21,22,23,24,25,26,27,28]. The number of patients included ranged from 10 [18] to 510 [10] (Table 1). The majority of studies were performed in the United States and all articles were retrospective cohorts (Appendix A). Date of publication of included studies ranged from 1986 [22] to 2023 [11].

## 3. Data Analysis

Continuous variables were displayed as mean or median values, depending on the metric used by the study. Categorical variables were reported as proportions, which we calculated by dividing the total events of interest by the overall at-risk population. Weighted means for both continuous and categorical values were calculated in order to adjust for the sample size of each study. All analyses were performed using Microsoft Excel version 16.73 (Microsoft Corp., Albuquerque, NM, USA) and StataSE 14 (StataCorp., College Station, TX, USA).

## 4. Results

### 4.1. What Are the Demographic and Clinical Characteristics of Patients with RIS?

A total of 1371 patients with RIS were included with a mean age of 58 at diagnosis, ranging from 28.7 [25] to 67 [20], a male: female ratio of 0.57 and a follow-up ranging from 20.4 [20] to 62.9 months [10]. The mean latency between the primary tumor and the RIS was 14.1 years and ranged from 8.6 [21] to 55.5 years [7]. The mean radiation dose received for the primary malignancy was 29.2 Gy and ranged from 10 [5] to 50.4 Gy [27]. The most common location was the trunk (59%), followed by extremities (21%) and pelvis (11%) (Table 2). The mean tumor size was 6.13 cm and 64% of RISs were high-grade. Regarding RIS histology, the most common histologic subtypes were undifferentiated pleomorphic sarcoma (UPS) and angiosarcoma (AS), occurring in 42.2 and 25.6% of cases, respectively (Figure 2).

### 4.2. What Are the Most Common Treatment Strategies of Patients with RIS?

Surgical resection was conducted in 68% of patients, with values ranging from 45% [25] to 100% [5] (Table 3). The majority of patients (74%) with appendicular RIS underwent limb-salvage surgery and 26% of them had an amputation as the primary procedure. Negative margins were reported in 58% of patients, with the rate ranging from 31% [3] to 90% [18]. QT, both neoadjuvant and adjuvant, was administered to 29% of patients, with values ranging from 0% [18] to 88% [3]. RT for the management of RIS was only described in 10 studies and it was performed in 15% of patients, with values ranging from 0% [12,18] to 53% [11].

### 4.3. What Are the Oncologic Outcomes of Patients with RIS?

The five-year overall survival was 45% among included studies, with values ranging from 13% [7] to 68% [25] (Table 4). At the last follow-up, 39% of patients had local recurrence, with values ranging from 20% [11] to 65% [12]. Metastasis occurred in 27% of patients with RIS, with reported rates ranging from 5% [8] to 67% [7].

## 5. Discussion

RISs represent an important long-term complication of radiation therapy and patients display worse outcomes than those with de novo STS when compared with historical controls. In our study, we found that RIS occurred most commonly in the trunk usually more than a decade after irradiation of the primary tumor. The most common histologic subtype was UPS (42.2%). Mainstay treatment consisted of surgical resection, with less than one third of patients receiving RT or QT. At five years, less than half of the patients (45%) were alive and 39% and 27% developed local recurrence and metastasis, respectively. To the best of our knowledge, this study represents the first systematic review performed to assess the oncologic outcomes of RIS located in the extremities or trunk.

Our study has several limitations. First, due to the rarity of this entity, we included studies with a small number of patients. Moreover, some of the included studies combined data of radiation-induced STSs and bone sarcomas or included locations other than the trunk and limbs in the analysis. Additional cases may have been missed in cohorts that focused on soft tissue sarcomas but did not make the distinction between de novo STS and RIS. Second, additional variables such as the type of primary tumor, chemotherapy regimen, radiotherapy scheme, disease-specific survival, time to local recurrence and time to metastasis were not available in the majority of included studies. Although this limitation is inherent to most systematic reviews, it did limit the depth of our analysis. Third, publication bias may have been present, with studies reporting poorer oncologic outcomes being less likely to be published. Fourth, our analysis was restricted to RIS and did not compare outcomes with a controlled cohort of patients with de novo STS due to the nature of the published literature. This might limit the generalizability of our findings.

In our study, the mean patient age at the time of ST-RIS diagnosis was 58.4 years, and 54% of patients were women. A female predominance in RIS has been previously described and is attributed to the higher incidence of breast cancer, usually treated with RT, in this population [8,20]. The mean latency time that we found was 14.1 years. It is hypothesized that latency period may be linked to both radiation dose and type of sarcoma [26]. Wiklund et al. stated that there is an inverse relationship between radiation dose and latency time, implying a threshold dose for developing RIS [8]. Conversely, other authors have suggested that higher dosages of RT might result in longer latency period times. This is based on the idea that the malignant transformation of cells is directly proportional to RT dosage up to a certain point, after which RT reduces the number of at-risk cells. Thus, the risk of RIS reaches a peak and then decreases as the dosage keeps increasing [22,29].

The most common histological type of RIS was UPS. This is important as the histological STS subtype carries prognostic value in terms of overall and disease-specific survival [5]. Moreover, the link between histologic subtype and prognosis has also been demonstrated in RIS. Studies have reported that UPS and malignant peripheral nerve sheath tumor (MPNST) RIS have worse oncological outcomes compared to leiomyosarcoma, fibrosarcoma and myxofibrosarcoma [5]. In addition, there may be a link between latency time and histologic subtypes [5].

The optimal treatment strategy for RIS remains unknown, Due to its locations within a previously irradiated field, the therapeutic alternatives for RIS are more limited in comparison with de novo STS [4]. In our study, approximately two thirds (68%) of patients with RIS underwent surgery and 58% of them attained negative margins. Although wide resection is currently considered the cornerstone of RIS management, this procedure has a lower likelihood of resecting the tumor with negative margins compared to de novo STS [5,20]. A previous study by Callese et al. reported that R0 margins after the resection of an RIS ranged from 31 to 91% [20]. The ample variability in R0 margin rates can be explained by differences in patient selection for surgical management [20]. Moreover, soft tissue fibrosis caused by prior RT may further complicate the surgical approach, making tissue plane identification, the accurate determination of the tumor extent and surgical field exposure difficult [4]. In addition, soft tissue fibrosis can decrease the effects of chemotherapy [4]. In our study, only 29% of patients with RIS received chemotherapy, lower than the rates reported in the literature for de novo STS. This pattern was not, however, seen in all studies included: Gladdy et al. reported that chemotherapy was used at a similar rate in both de novo STS and RIS [5]. It is important to highlight both the risks and benefits of chemotherapy, especially considering the large proportion of elderly patients and the fact that, historically, it has been mainly used for palliative care [26]. Novel discoveries and benefits in local control should incline physicians to use chemotherapy beyond palliative care and as a potentially useful tool for RIS management.

In our study, only 15% of patients received RT to treat the RIS, highlighting the reluctance of radiation oncologists to reirradiate these tumors. RT as a part of the management of RIS has been traditionally rejected due to concerns about increased toxicity and complications from tissue reirradiation [4]. Likewise, reirradiation can interfere with adequate margin resection, given that irradiated tissue has fibrotic changes that could be confused with tumoral tissue and vice versa. Riad et al. suggested that patients with locally RIS could benefit from reirradiation [4]. To consider reirradiation, adequate patient selection is critical and should consider the surgical margins, previous radiation dosage (Gy), volume of normal tissue irradiated, estimated normal tissue recovery after RT and presence of adjacent critical radiosensitive structures. Current RT technologies, such as image-guided radiotherapy (IMRT), brachytherapy or proton therapy, offer the ability to safeguard normal tissue while precisely targeting the affected area, providing greater flexibility in incorporating RT as a fundamental component of the management of RIS [4,20].

Outcomes in patients with RIS have been reported to be worse in comparison with sporadic STS [10,20,27]. In our study, the five-year overall survival was 45%, similar to what was found by Gladdy et al. (41%) and Kim et al. (45%) [5,27]. In contrast, Italiano et al. reported a higher five-year overall survival (51.9%). Their population, however, only included patients who received surgical management and obtained R0/R1 margins. Based on their results, they advocated for applying the same surgical curative treatment intent in RIS as in de novo sarcomas [10]. Even though the latest article reported a better outcome, still, when compared historically with de novo soft tissue sarcomas, the outcomes are inferior. The persistent poor overall survival in RIS, despite a recent slight increase reported by Italiano et al., can be attributed to the intrinsic histopathological characteristics of RISs, their limited response to adjuvant treatment and the higher rates of local recurrence and metastasis compared to de novo STS.

Our study found a metastasis rate of 28%, which is lower than the 50–70% rate for de novo high-grade STS [30,31,32]. In contrast, Callesen et al. reported a higher metastasis rate at the time of diagnosis in patients with RIS (22% compared to 10% for de novo STS) [20,32]. This difference can be attributed to the longer latency period and the more aggressive nature of the RIS tumor [20,26]. We also observed a local recurrence rate of 38%, lower than previously reported by other studies (ranging from 41 to 65%) [7,12,21], but higher than the 26% rate reported by Riad et al. [4]. It is worth noting that both our findings and those described in the literature of RIS are significantly higher than the local recurrence of 6.5 to 9% reported for de novo STS [4,33,34,35]. Notably, lower local recurrence rates have been observed in patients with RIS who underwent reirradiation (7.7%), compared to patients who only received surgical management (34.5%) [4].

## 6. Conclusions

Our study represents the first and largest systematic review performed to assess outcomes of patients with RIS. The optimal treatment strategy for this rare entity remains unknown, yet expanding treatment strategies beyond surgical resection is supported by more recent studies. In our study, the most common treatment was surgical resection, with RT and chemotherapy being administered in less than one third of patients. Patients with RIS exhibited poor oncologic outcomes, with the five-year overall survival below 50% and metastasis rate at 27%. Local recurrence was reported in 39% of patients. Future studies should evaluate the efficiency of combined treatment strategies on this very rare and unique group of patients. Moreover, studies should compare RIS with de novo STS while controlling for confounders to have additional insight on the biologic behavior of these tumors.

## Figures and Tables

**Figure 1 cancers-15-05584-f001:**
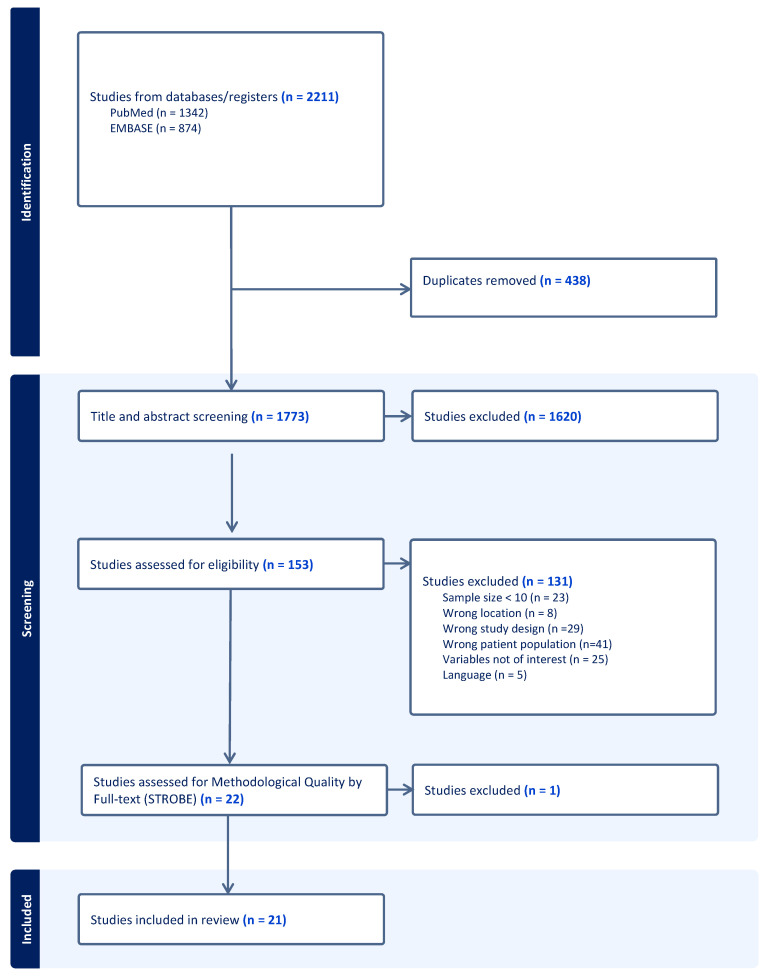
PRISMA flowchart for our literature search and selection of relevant articles.

**Figure 2 cancers-15-05584-f002:**
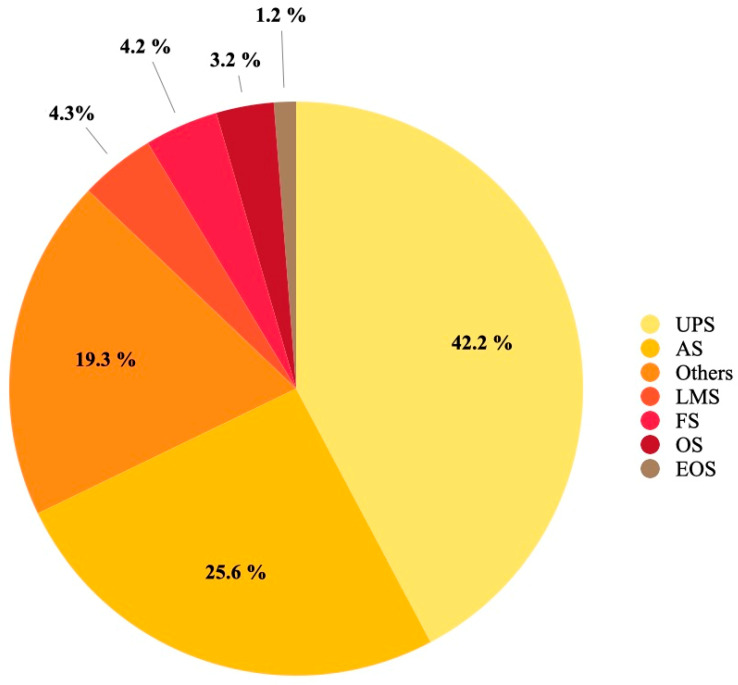
Pie chart of the histology distribution of the included RISs. UPS: undifferentiated pleomorphic sarcoma; AS: angiosarcoma; LMS: leiomyosarcoma; FS: fibrosarcoma; OS: osteosarcoma; EOS: extraskeletal osteosarcoma; others: malignant peripheral nerve sheath tumor, pleomorphic leiomyosarcoma, myxofibrosarcoma, Ewing sarcoma, malignant schwannoma, rhabdomyosarcoma, hemangiosarcoma, lymphangiosarcoma.

**Table 1 cancers-15-05584-t001:** Demographic characteristics of included patients.

Author	RIS (*n*)	Age (Years) ^α^	Sex (M:F)	Latency (Years) ^α^	Dose (Gy) ^α^	FU (Months) ^α^
Kao et al. [11]	57	60 ^+^	0.57 ^+^	13 ^+^		28.3 ^+^
Spalek et al. [3]	49	57 ^+^	2.87 ^+^		11 ^ϕ+^	
Callesen et al. [20]	49	67 ^+^	0.21 ^+^	11 ^ϕ+^		20.4 ^ϕ+^
Italiano et al. [10]	510	66	0.24 ^+^			62.9
Joo et al. [26]	19	56	0.58 ^+^	19.5	43.4 ^ϕ^	25 ^ϕ^
Dineen et al. [23]	55	59 ^ϕ+^	0.67 ^+^	9.33 ^ϕ+^	50 ^ϕ+^	
Kim et al. [27]	12	48	0.20	12.1 ^ϕ+^	50.4 ^ϕ+^	23.1 ^ϕ+^
Riad et al. [4]	44	56 ^ϕ+^	0.69	45 ^ϕ^	12.4	29 ^ϕ^
Gladdy et al. [5]	108	58.5 ^ϕ+^	0.73	10 ^+^	10 ^+^	26.7 ^ϕ^
Neuhaus et al. [12]	54	58 ^ϕ+^		11 ^ϕ+^	50 ^ϕ+^	53 ^ϕ+^
Holt et al. [24]	38	33	0.73 ^+^	15.4	15.4	23
Thijssens et al. [7]	15	59	0.07	55.5	10.8	26.6
Cha et al. [21]	64	62 ^+^	0.86 ^+^	8.6 ^ϕ+^		36 ^ϕ+^
Fang et al. [6]	14	58.4	0.08	12.6	12.6	24
Lagrange et al. [9]	56	55.5	0.27	13.3	13.3	38.8
Inoue et al. [25]	11	28.7 ^+^	0.59 ^+^	17 ^+^		
Bloechle et al. [18]	10	51.1	0.11	15.8	39.9	56.4
Brady et al. [19]	113	41 ^+^	0.90 ^+^	10.3 ^+^	50 ^ϕ+^	37 ^+^
Wiklund et al. [8]	20	63	0.10		13.5	44.4
Laskin et al. [28]	53	51	0.39	9.6	38.6	23.6
Davidson et al. [22]	20	35.5		16.8 ^ϕ+^	37.3	

FU: follow-up; RIS: radiation-induced soft tissue sarcoma. ^α^ values in this column refer to the mean. ^ϕ^ values in these cells refer to the median. ^+^ calculated from entire study sample and not restricted to RIS located in the appendicular skeleton, pelvis or trunk.

**Table 2 cancers-15-05584-t002:** Clinical characteristics of included patients.

Author	Location			High Grade (%)	Size (cm) ^α^	Histology
	Trunk (%)	Extremity (%)	Pelvis (%)			
Kao et al. [11]	82%	18%	0%	66% ^+^	7.4 ^+^	UPS (45%), AS (23%), MPNST (14%), LMS (12%), RS (4%)
Spalek et al. [3]	57% ^++^	14% ^++^	29% ^++^	95%		UPS (30%), AS (14%), MFS (12%), MPNST (8%), other (24%)
Callesen et al. [20]	57%	13%	30%	73%^+^	8 ^ϕ+^	UPS (48%), AS (25%), Ewing sarcoma (11%), LMS (6%) ^+^
Italiano et al. [10]	58%	16%	10%	43%	5.5	AS (38%), UPS (34%), other (28%)
Joo et al. [26]	21%	26%	26%	94%	5.7	FS (42%), AS (17%), UPS (17%), other (24%)
Dineen et al. [23]	76%	24%	0%		6 ^ϕ^	UPS (100%)
Kim et al. [27]					4.8 ^ϕ+^	UPS (33%), AS (6%), CS (3%), other (58%) ^+^
Riad et al. [4]	20%	80%	0%	70%	7.1	UPS (36%), AS (18%), LS (9%), other (37%)
Gladdy et al. [5]	74%	26%	0%	83%^+^	5.7^+^	UPS (34%), AS (21%), LMS (12%), FS (12%)
Neuhaus et al. [12]	74%	9%	17%	87%^+^	8^+^	LMS (28%), UPS (16%), AS (13%) ^+^
Holt et al. [24]						UPS (36%), OS (28%), LMS (6.4%), RMS (4.3%)
Thijssens et al. [7]	80%	0%	20%	40%		AS (40%), UPS (40%), FS (13%), pleomorphic LMS (7%)
Cha et al. [21]	60%	40%	0%	79%^+^	≤5 cm: 40% >5–10 cm: 37% >10 cm: 23%	UPS (23%), FS (15%), AS (15%), LMS (12%) ^+^
Fang et al. [6]	50%	14%	36%	100%	7.8	UPS (50%), EOS (43%), FS (7%)
Lagrange et al. [9]	46%	10%	16%	55% ^+^		UPS (43%) AS (12%) OS (9%), FS (11%) ^+^
Inoue et al. [25]	0%	27%	73%	74% ^+^	9 ^+^	FS (62%), UPS (25%), other (13%) ^+^
Bloechle et al. [18]	50%	40%	10%	50%	6.2	UPS (60%), HS (20%)
Brady et al. [19]				87% ^+^	6 ^+^	OS (21%), UPS (16%), AS/LA (15%) ^+^
Wiklund et al. [8]	50%	15%	35%	100%	8.8	UPS (30%), EOS (20%), FS (20%), LMS (15%), MS (5%), AS (5%)
Laskin et al. [28]	69%	17%	14%			UPS (77%), FS (9%), EOS (6%), MS (6%), Others (2%) ^+^
Davidson et al. [22]						UPS (40%), FS (20%), LA (10%), EOS (10%)

AS: angiosarcoma; CS: chondrosarcoma; EOS: extraskeletal osteosarcoma; FS: fibrosarcoma; HS: hemangiosarcoma; LA: lymphangiosarcoma; LMS: leiomyosarcoma; LS: liposarcoma; MS: malignant schwannoma; MPNST: malignant peripheral nerve sheath tumor; OS: osteosarcoma; RMS: rhabdomyosarcoma; UPS: undifferentiated pleomorphic sarcoma. ^α^ values in this column refer to the mean; ^ϕ^ values in these cells refer to the median. ^+^ calculated from entire study sample and not restricted to RIS located in the appendicular skeleton, pelvis or trunk; ^++^ value inferred from proportions.

**Table 3 cancers-15-05584-t003:** Treatment strategies of included patients.

Author	Surgery	Type of Surgery		Negative Surgical Margins	(Neo)-Adjuvant RT	(Neo)-Adjuvant QT
		Limb-Salvage Surgery	Amputation			
Kao et al. [11]	66% ^+^			43% ^+^	53% ^+^	47% ^+^
Spalek et al. [3]	67% ^+^			31% ^+^	33%	88% ^+^
Callesen et al. [20]	78% ^+^			80% ^+^	9% ^+^	9% ^+^
Italiano et al. [10]	53%				9%	31%
Joo et al. [26]	94%	95%	5%	74% ^+^		52%
Dineen et al. [23]						
Kim et al. [27]	100%				16%	25%
Riad et al. [4]	95%	86%	14%	83%	30%	18%
Gladdy et al. [5]	100% ^+^			69% ^+^	22% ^+^	18%
Neuhaus et al. [12]	72% ^+^			75% ^+^	0%	9% ^+^
Holt et al. [24]						
Thijssens et al. [7]	67%	40%	60%	60%	7%	13%
Cha et al. [21]	90% ^+^			47% ^+^		20% ^+^
Fang et al. [6]	93%					
Lagrange et al. [9]						
Inoue et al. [25]	45% ^+^	20%	80%			
Bloechle et al. [18]	100%	90%	10%	90%	0%	0%
Brady et al. [19]				42% ^+^		
Wiklund et al. [8]						
Laskin et al. [28]						
Davidson et al. [22]						

QT: chemotherapy; RT: radiation therapy. ^+^ calculated from entire study sample and not restricted to RIS located in the appendicular skeleton, pelvis or trunk.

**Table 4 cancers-15-05584-t004:** Oncologic outcomes of included patients.

Author	5y OS	Local Recurrence (% of Total)	Metastasis Rate (% of Total)
Kao et al. [11]	42% ^+^	20%	9%
Spalek et al. [3]	17%	31% ^+^	28% ^+^
Callesen et al. [20]	32% ^+^		22% ^+^
Italiano et al. [10]	53%		
Joo et al. [26]	50%	26%	
Dineen et al. [23]	47%	58%	
Kim et al. [27]	45% ^+^	47% ^+^	37% ^+^
Riad et al. [4]	44%	26%	50%
Gladdy et al. [5]			
Neuhaus et al. [12]	45% ^+^	65% ^+^	44% ^+^
Holt et al. [24]	51%		
Thijssens et al. [7]	13%	40%	67%
Cha et al. [21]	41% ^+^	47% ^+^	28% ^+^
Fang et al. [6]	53%	30%	20%
Lagrange et al. [9]	35%	30%	52%
Inoue et al. [25]	68% ^+^		
Bloechle et al. [18]	40%		10%
Brady et al. [19]	41% ^+^		8% ^+^
Wiklund et al. [8]	30%		5%
Laskin et al. [28]	37%		
Davidson et al. [22]	14%		20%

^+^ calculated from entire study sample and not restricted to RIS located in the appendicular skeleton, pelvis or trunk.

## Data Availability

The data presented in this study are available in this article (and Appendix A).

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
