# Peer review of "Treatment and Outcomes of Radiation-Induced Soft Tissue Sarcomas of the Extremities and Trunk—A Systematic Review of the Literature"

_cancers, 2023, doi:10.3390/cancers15235584_

Round 1

Reviewer 1 Report

Comments and Suggestions for Authors

The aim of this systematic review was to analyze how RIS is treated, and the outcomes patients face. 

The topic is interesting. The study is well designed. However, it is very possible that some cases have been missed because of the search strings used. It is possible that some series did not focus on radio induced sarcomas but included some cases of those. Please acknowledge as a limitation.

Fig 1 low quality.

Table 1. please specify the total number of STS in the studies.

Please define a radio induced sarcoma. 

Treatments details should be detailed further.

Also, please provide details about LR rate.

Discussion: I would further discuss about differences with non radio induced STS and with prognosis of bone radio induced sarcomas.

Author Response

Reply to Reviewer 1 

  1. The aim of this systematic review was to analyze how RIS is treated, and the outcomes patients face. The topic is interesting. The study is well designed. However, it is very possible that some cases have been missed because of the search strings used. It is possible that some series did not focus on radio induced sarcomas but included some cases of those. Please acknowledge as a limitation.

The authors appreciate this recommendation. We have included it as a limitation to our study. (See lines 210-212)

  1. Fig 1 low quality.

The authors appreciate this observation and have improved the quality of the figure (see line 123). Additionally, we have attached a high-quality to the submission file, which can be used during the final manuscript production phase.

  1. Table 1. please specify the total number of STS in the studies.

The authors appreciate this recommendation. The total number of radiation-induced soft tissue sarcomas in each study is described in the second column of table 1 described as “RIS (n)”.  In the table 1. legend and in the text, we describe the acronym “RIS” as radiation-induced soft tissue sarcoma. (See line 132)

  1. Please define a radio induced sarcoma.

The authors appreciate this recommendation. We defined radio-induced sarcoma based on the diagnostic criteria published by Cahan et al in 1948. (See lines 46-50)

  1. Treatments details should be detailed further.

The authors appreciate this recommendation. We have expanded and clarify our treatment strategies definitions. (See lines 92-97)

  1. Also, please provide details about LR rate.

The authors appreciate this recommendation. We have included a definition of local recurrence rate. (See lines 102-103 and 166).

  1. Discussion: I would further discuss about differences with non-radio induced STS and with prognosis of bone radio induced sarcomas.

The authors appreciate the recommendation. We have expanded our discussion regarding differences in prognosis of non-radiation induced STS and radiation induced STS. (See lines 278-282)  

Reviewer 2 Report

Comments and Suggestions for Authors

The manuscript by Inchastegui et al. analyses the literature on a rare, but very interesting disease for those involved in sarcoma care. The analysis was performed with thoroughness and was adequately described. Only one third of the patients underwent chemo and/or radiotherapy and the reasons for this are not described. If possible on the basis of the data present in the literature analysed, I would give more space to the topic, the time elapsed between first-line irradiation and the appearance of the RIS and the technological evolution of the last years would have allowed the use in a higher percentage of cases of a therapy that is certainly effective in local control.

The term 'de novo RIS' I think should be replaced by 'de novo STS'.

Author Response

Reply to Reviewer 2

  1. The manuscript by Inchaustegui et al. analyses the literature on a rare, but very interesting disease for those involved in sarcoma care. The analysis was performed with thoroughness and was adequately described. Only one third of the patients underwent chemo and/or radiotherapy and the reasons for this are not described. If possible, on the basis of the data present in the literature analyzed, I would give more space to the topic, the time elapsed between first-line irradiation and the appearance of the RIS and the technological evolution of the last years would have allowed the use in a higher percentage of cases of a therapy that is certainly effective in local control.

The authors appreciate this comment. The small percentage of patients receiving chemotherapy and/or radiotherapy has been further discussed. (See lines 249-252 and 255-263)

  1. The term 'de novo RIS' I think should be replaced by 'de novo STS'.

The authors appreciate this observation. We have corrected the term “de novo RIS” to “de novo STS”.  (See lines 40, 218, 240, 252, and 303)

Reviewer 3 Report

Comments and Suggestions for Authors

The authors presented  apapaer about "Treatment and Outcomes of Radiation-Induced Soft Tissue Sarcomas of the Extremities and Trunk – A Systematic Review of the Literature".

They used the correct methodology for a systematic review (PRISMA), they evaluated the quality (STROBE) and they obtained an addtional quality evaluation about the appropiateness of this review (PROSPERO).

They at investigating 3 main topics namely the demographic and clinical characteristics of patients with RIS, the most common treatment strategies of patients with RIS and the oncologic outcomes of patients with RIS.

In my opinion they managed to anwer all of the 3 topics adequately.

I compliment with the authors for their comprehensive work.

Author Response

Reply to Reviewer 3

  1. The authors presented  appear about "Treatment and Outcomes of Radiation-Induced Soft Tissue Sarcomas of the Extremities and Trunk – A Systematic Review of the Literature".They used the correct methodology for a systematic review (PRISMA), they evaluated the quality (STROBE) and they obtained an additional quality evaluation about the appropriateness of this review (PROSPERO).They at investigating 3 main topics namely the demographic and clinical characteristics of patients with RIS, the most common treatment strategies of patients with RIS and the oncologic outcomes of patients with RIS. In my opinion they managed to answer all of the 3 topics adequately. I compliment the authors for their comprehensive work.

The authors appreciate this comment.

Round 2

Reviewer 1 Report

Comments and Suggestions for Authors

The Authors made great efforts in the attempt to ameliorate their paper. it is now suitable for publication.